# Relationship between Amino Acid Metabolism and Bovine In Vitro Follicle Activation and Growth

**DOI:** 10.3390/ani13071141

**Published:** 2023-03-23

**Authors:** Kenichiro Sakaguchi, Kohei Kawano, Yuki Otani, Yojiro Yanagawa, Seiji Katagiri, Evelyn E. Telfer

**Affiliations:** 1Institute of Cell Biology, School of Biological Sciences, College of Science and Engineering, University of Edinburgh, The Hugh Robson Building, 15 George Square, Edinburgh EH8 9XD, UK; 2Laboratory of Theriogenology, Department of Clinical Sciences, Faculty of Veterinary Medicine, Hokkaido University, Kita 18 Nishi 9, Kita-ku, Sapporo 060-0818, Japan; 3Laboratory of Anatomy, Department of Basic Veterinary Sciences, Faculty of Veterinary Medicine, Hokkaido University, Kita 18 Nishi 9, Kita-ku, Sapporo 060-0818, Japan

**Keywords:** amino acid, bovine, in vitro growth, primordial follicle, secondary follicle

## Abstract

**Simple Summary:**

Bovine ovaries at all ages contain high numbers of immature eggs (oocytes) contained in follicles, but only a small proportion will ever be ovulated, with the rest destined to degenerate. In vitro growth (IVG) is a culture technique to support the development of immature oocytes/follicles in vitro. Bovine primordial follicles can be grown in vitro to the antral stage, but further optimization of the culture system is required to support development to maturity and fertilization. The present study focuses on the amino acid metabolism of early-stage bovine follicles during IVG to determine whether this can be correlated with development to provide a non-invasive marker of follicle health in vitro. The results indicate possible candidate amino acids and metabolites as potential markers of health status for in vitro-grown follicles.

**Abstract:**

The amino acid metabolism of bovine follicles during in vitro growth (IVG) was evaluated to identify potential indicators of health during culture. The bovine ovarian cortex was sliced, prepared as strips, and cultured for 6 days. Tissue samples were examined histologically before and after 6 days of culture, and the degree of follicle activation was classified as either high or low based on the number of growing secondary follicles present (high: 7~11; low: 0~1). In a separate experiment, secondary follicles (diameter range: 100~200 μm) were manually isolated and cultured, and their growth was monitored for 6 days. Cultured follicles were classified as growth or degenerate based on diameter change during culture (growth: +60.5~74.1 μm; degenerate: −28~15.2 μm). Free amino acids and their metabolites were measured in the spent culture medium from each group. In cultured ovarian cortical strips, the concentration of α-aminoadipic acid was significantly higher in the low activation group than in the high group (*p* < 0.05), while those of methionine, lysine, and arginine were higher in the high activation group. In cultured isolated secondary follicles, concentrations of methionine, tyrosine, histidine, and hydroxyproline were higher in the degenerate group (*p* ≤ 0.05). In conclusion, amino acid metabolism has the potential to serve as an indicator of primordial follicle activation and subsequent growth rate during bovine IVG.

## 1. Introduction

In most mammals, female germ cells (oocytes) are formed within the ovary before or shortly after birth. Oocytes, arrested at the dictyate stage of Prophase I, become enclosed by somatic (granulosa) cells to form primordial follicles, which are quiescent until activated to grow. The mammalian female is born with thousands of primordial follicles that are activated to grow throughout life (primordial follicle activation) [1,2]. However, over 99% of primordial follicles are destined to die, with only a small proportion ever reaching the ovulatory stage. 

The ability to develop primordial follicles in vitro would have advantages in fertility preservation for women as well as animal production and the conservation of rare breeds [3]. Techniques of in vitro growth (IVG) to support the activation and growth of immature oocytes from follicles at early developmental stages to acquire competence to undergo meiotic maturation, be fertilized, develop to the blastocyst stage, and produce offspring have been developed. In mice, pups have been produced from oocytes derived from primordial follicles grown entirely in vitro. This methodology utilized a combination of the organ culture of neonatal ovaries and the culture of oocyte-cumulus-granulosa complexes (OCGCs) to obtain mature oocytes that could be fertilized [4,5]. Similar systems have been developed in humans, where metaphase II oocytes can be obtained from primordial follicles grown in a multi-step culture system following in vitro maturation (IVM) [6]. In addition, a 2-step culture system for bovine primordial follicles has been established, and antral follicles and oocytes with diameters measuring over 100 μm, which are close to maturity, can be obtained [7]. However, there have been no reports of embryos or mature oocytes in cattle from using IVG [8]. Therefore, further improvement of the cultural system is required.

Metabolomics is a powerful method that permits the analysis of various metabolites in a biological system, enabling metabolic changes related to disturbances of homeostasis to be detected [9]. Among the various metabolites that can be monitored, amino acids are the most suitable candidates for focused metabolomics since they play vital physiological roles as both substrates and regulators in several metabolic pathways [10,11,12]. Free amino acids in culture media have diverse roles such as protein synthesis, energy sources, osmotic pressure, and pH adjustment, while toxic ammonia is produced by natural degradation [13]. The analysis of free amino acids in a spent culture medium has the potential as a non-invasive method to predict oocyte developmental competence in IVM [14] and embryo culture [15,16,17,18]. Therefore, the aim of this study is to examine the relationship between free amino acids in culture media and the developmental potential of follicles grown in vitro. In the present study, our objective was to identify amino acid species and metabolites indicative of follicular growth competence in bovine primordial and secondary follicles grown in vitro.

## 2. Materials and Methods

### 2.1. Chemicals

Unless otherwise stated, all chemicals used in this study were purchased from Sigma-Aldrich (St. Louis, MO, USA).

### 2.2. Preparation of Strips of Ovarian Cortex Containing Primordial Follicles 

The collection and preparation of ovarian tissue have been previously described [7,19]. Ovaries of Holstein cows were obtained from a local abattoir and transported in HEPES (25 mM)-buffered tissue culture medium 199 supplemented with amphotericin B (2.5 μg/mL), sodium pyruvate (25 μg/mL), penicillin G (75 μg/mL), and streptomycin (50 μg/mL) pre-warmed at 37°C. Upon arrival at the laboratory, fine cortical strips were removed from the ovaries using a surgical blade (no. 24, FEATHER Safety Razor Co., Ltd., Osaka, Japan), and then transferred into HEPES (25 mM)-buffered Leibovitz medium (Thermo Fisher Scientific, Waltham, MA, USA) supplemented with sodium pyruvate (2 mM), glutamine (2 mM), bovine serum albumin (BSA, Fraction V, 3 mg/mL), penicillin G (75 μg/mL), and streptomycin (50 μg/mL) (dissection medium) pre-warmed at 37 °C. 

Excess stromal tissue was trimmed using forceps and a scalpel blade. The tissue was gently stretched using the blunt edge of a scalpel blade with the cortex uppermost and cut into small strips sized 4 mm × 2 mm × 1 mm. To ensure the strips contained predominantly uni-laminar follicles, any follicles measuring >40 μm were removed by dissection. Three replicate cultures were established with 10–12 cortical strips in each. Cortical strips were cultured individually in 24-well cell culture plates (Corning Life Sciences, Tewksbury, MA, USA) with 300 μL of HEPES (25 mM)-buffered McCoy’s 5a medium (Thermo Fisher Scientific, Waltham, MA, USA) supplemented with glutamine (3 mM; Thermo Fisher Scientific, Waltham, MA, USA), BSA (Fraction V, 0.1%), penicillin G (0.1 mg/mL), streptomycin (0.1 mg/mL), transferrin (2.5 μg/mL), selenium (4 ng/mL), insulin (10 ng/mL), sodium ascorbic acid (50 μg/mL), and recombinant human follicle-stimulating hormone (FSH, 1 ng/mL; R&D systems, Minneapolis, MN, USA). Strips were cultured for 6 days at 39 °C in humidified air with 5% CO_2_. The remaining strips were fixed in 10% neutral buffered formalin for approximately 24 h. Every second day of culture, half (150 μL) of the culture medium was replaced with the same amount of fresh medium. The spent media collected on days 2, 4, and 6 of the culture were stored at −80 °C until free amino acid analysis was conducted. After 6 days of culture, cortical strips were fixed in 10% neutral buffered formalin for 24 h.

### 2.3. Histological Analysis

Bovine ovarian cortical strips on days 0 (n = 11) and 6 (n = 32) were fixed in 10% neutral buffered formalin for approximately 24 h. After embedding in paraffin, the strips were cut into 5 μm thick sections. The sections were deparaffinized and stained with hematoxylin and eosin. Analysis of follicles was performed on every section under the light microscope at ×400 magnification. Follicle developmental stages were categorized using an established system [20] (Figure 1). The number of follicles at each stage was recorded for days 0 and 6 of the culture. 

### 2.4. Isolation and Culture of Secondary (Preantral) Follicles

The bovine ovarian cortex was sliced into thin strips (thickness <1 mm) using a surgical blade (no. 24, FEATHER) and transferred into a dissection medium pre-warmed at 37 °C. Secondary follicles (100–200 μm) were mechanically dissected using a surgical blade (no. 15, FEATHER), and those with an intact basement membrane and no antral cavity were selected for culture (n = 60) (Figure 2). Three replicate cultures were established, with 13–24 secondary follicles in each. Selected secondary follicles were placed individually on 96-well V-bottomed culture plates (Corning Life Sciences, Tewksbury, MA, USA) in 150 μL of HEPES (25 mM)-buffered McCoy’s 5a medium (Thermo Fisher Scientific, Waltham, MA, USA) supplemented with glutamine (3 mM; Thermo Fisher Scientific, Waltham, MA, USA), BSA (Fraction V, 0.1%), penicillin G (0.1 mg/mL), streptomycin (0.1 mg/mL), transferrin (2.5 μg/mL), selenium (4 ng/mL), insulin (10 ng/mL), sodium ascorbic acid (50 μg/mL), recombinant human FSH (1 ng/mL), and recombinant human activin-A (100 ng/mL; R&D systems, Minneapolis, MN, USA). The follicles were cultured individually for 6 days at 39 °C in humidified air with 5% CO_2_. Every second day of culture, half (75 μL) of the culture medium was replaced with the same amount of fresh medium. The spent media collected on days 2, 4, and 6 of the culture were stored at −80°C until preparation for free amino acid analysis. 

### 2.5. Follicle Diameter Measurement

At the start and end of the culture, follicles were photographed under a stereo microscope at ×50 magnification with an attached CCD camera (Moticam 2000, Shimadzu Rika Corporation, Tokyo, Japan). Follicle diameters were measured and calculated as the mean value of the longest and shortest diameters using software (Motic Images Plus 2.2s, Shimadzu Rika Corporation). Follicles that ruptured and those that could not be measured because of stromal overgrowth were excluded from the study.

### 2.6. Free Amino Acid Analysis

Free amino acids in spent culture media were analysed using the post-column colorimetric derivatization with the ninhydrin method using a High Speed Amino Acid Analyzer (L-8900; Hitachi High-Tech Corp., Tokyo, Japan). It can analyse 41 molecular species (amino acids, peptides, amino sugars, and amino alcohols), which are leucine (Leu), isoleucine (Ile), valine (Val), threonine (Thr), tryptophan (Trp), phenylalanine (Phe), methionine (Met), lysine (Lys), histidine (His), asparagine (AspNH_2_), aspartic acid (Asp), alanin (Ala), arginine (Arg), glycine (Gly), glutamine (GluNH_2_), glutamic acid (Glu), proline (Pro), hydroxyproline (Hypro), serine (Ser), tyrosine (Tyr), cystine (Cys), phosphoserine (P-Ser), taurine (Tau), phosphoethanolamine (PEA), urea, sarcosine (Sar), α-aminoadipic acid (a-AAA), citrulline (Cit), α-aminobutyric acid (α-ABA), cystathionine, β-Alanine (β-Ala), β-aminoisobutyric acid (β -AiBA), γ-Aminobutyric acid (γ-ABA), ethanolamine (EOHNH_2_), ammonia (NH_3_), hydroxylysine (Hyl), ornithine (Orn), 1-methylhistidine (His(1-Me)), 3-methylhistidine (His(3-Me)), carnosine, and anserine.

Samples selected to be analyzed were diluted ×3 with 0.02M Hydrochloric acid. Since Glutamine was high within the culture medium, this was measured in a sample that was diluted ×30. 

### 2.7. Experimental Design

#### 2.7.1. Experiment 1 Relationship between Amino Acid Metabolism and Primordial Follicle Activation In Vitro 

Samples from cortical strip (primordial) culture for free amino acid analysis were selected on the basis of whether the strip showed high or low activation of primordial follicle growth (Figure 3). The criteria for high activation were that the strips contained predominantly primordial follicles and a maximum of 2 secondary follicles before culture and up to 11 secondary follicles after 6 days of culture, whereas the low activation group showed less than 2 secondary follicles present after 6 days in culture. The top three samples, defined by the number of secondary follicles after culture (11, 10, and 7 secondary follicles), were clustered as the high activation group, and the bottom three samples (0, 0, and 1 secondary follicles) as the low activation group (Figure 3). The spent culture medium on days 2, 4, and 6 from these samples was subjected to free amino acid analysis. The concentration of free amino acids and their metabolites were compared between the groups (low activation vs. high activation) and culture periods (day 2 vs. day 4 vs. day 6).

#### 2.7.2. Experiment 2 Relationship between Amino Acid Metabolism and Secondary Follicle Growth In Vitro

The selection of samples from cultured secondary follicles for free amino acid analysis was based on changes in follicle diameter measurements during culture. Follicles were measured at the start of culture and at days 2, 4, and 6 as previously described [21]. The measurement at day 0 was subtracted from those on day 6 to obtain a measurement of change. As described in Figure 4, 25 follicles showed a significant increase in size during culture, while 4 follicles decreased in size. The top three samples in the growth of secondary follicles during culture (74.1, 64.7, and 60.5 μm) were clustered as the growth group, and the bottom three samples, which showed an overall decrease in size (−28.0, −17.0, and −15.2 μm), were defined as the degenerate group. The spent culture medium from these two groups collected on days 2, 4, and 6 was subjected to free amino acid analysis. The concentration of free amino acids and their metabolites were compared between the groups (growth vs. degenerate) and culture periods (day 2 vs. day 4 vs. day 6). 

#### 2.7.3. Experiment 3 Difference in Amino Acid Metabolism between Cortical Strip Culture for Primordial Follicle Activation and Secondary Follicle Growth In Vitro

To compare cortical strip cultures with secondary follicle cultures, all results of free amino acid analysis from experiment 1 were pooled and compared to the pooled results from experiment 2. The result was compared between the types of culture (cortical strip culture vs. secondary follicle culture) and culture periods (day 2 vs. day 4 vs. day 6).

### 2.8. Statistical Analysis

All statistical analyses were performed using software (JMP Pro 14, SAS Institute, Cary, NC, USA). The ratio of follicles at each stage in cortical strip culture was compared between the low activation group and the high activation group using Student’s *t*-test or the Wilcoxon test. Diameters of secondary follicles before and after secondary follicle culture were compared between the groups designated growth and degenerate using Student’s *t*-test. Concentrations of free amino acids and metabolites were analyzed using a two-way analysis of variance (ANOVA). For the two-way ANOVA, we used the Fit Model platform in JMP Pro 14. The model included the effects of groups (low activation vs. high activation, growth vs. degenerate, or cortical strip culture vs. secondary follicle culture), days of culture (day 2 vs. day 4 vs. day 6), and their interactions. Student’s *t*-test or Tukey–Kramer’s honestly significant difference (HSD) test was used as post hoc tests. 

## 3. Results

### 3.1. Relationship between Amino Acid Metabolism and Primordial Follicle Activation In Vitro 

As shown in Figure 5, the ratio of primordial follicles was significantly higher in the low activation group than the high activation group on day 6 (*p* < 0.05), while growing follicles (primary follicles and secondary follicles) after culture were higher in the high activation group than the low activation group (*p* < 0.05).

As shown in Table 1, days of culture affected 10 essential amino acids (Arg, His, Ile, Leu, Lys, Met, Phe, Thr, Trp, Val), 3 proteinogenic non-essential amino acids (Gly, Pro, Tyr), 4 non-proteinogenic amino acids (α-AAA, Hylys, Hypro, Orn), and PEA (*p* < 0.05). Groups affected 3 essential amino acids (Arg, Lys, Met) and α-AAA (*p* < 0.05). During culture, concentrations of 10 essential amino acids (Arg, His, Ile, Leu, Lys, Met, Phe, Thr, Trp, Val), 3 proteinogenic non-essential amino acids (Gly, Pro, Tyr), 3 non-proteinogenic amino acids (Hylys, Hypro, Orn), and EOHNH_2_ increased (all concentrations are shown in Appendix A, *p* < 0.05), while the concentration of α-AAA and PEA decreased regardless of the groups (Appendix A, *p* < 0.05). 

The molecular species that showed differences between groups are shown in Figure 6. Arg and Lys were significantly higher (*p* < 0.05) in the high activation group whilst α-AAA was higher in the low activation group on day 2 of culture (*p* < 0.05). The overall average concentrations of Arg, Lys, and Met at each sampling time were higher (*p* < 0.05) in the high activation group (*p* < 0.05), while the overall average concentrations of α-AAA were higher in the low activation group (*p* < 0.05).

### 3.2. Relationship between Amino Acid Metabolism and Secondary Follicle Growth In Vitro 

As shown in Figure 7, the diameters of both groups were similar before the culture, while the growth group showed a significant increase in size during the culture period compared to the degenerate group (*p* < 0.05).

As shown in Table 2, there was an interaction between days of culture and groups in PEA. Days of culture affected 3 proteinogenic non-essential amino acids (Asp, Gly, and GluNH_2_), P-Ser, Urea, and NH_3_ (*p* < 0.05). Groups affected His (*p* = 0.0334), Met (*p* = 0.0528), Tyr (*p* = 0.0580), and Hypro (*p* = 0.0362). During culture, concentrations of 2 proteinogenic non-essential amino acids (Asp and Gly), P-Ser, Urea, and NH_3_ increased (Appendix A, *p* < 0.05), while the concentration of GluNH_2_ decreased regardless of the groups (Appendix A, *p* < 0.05). The molecular species that showed differences between the groups are shown in Figure 8. His and Met tended to be higher in the degenerate group than in the growth group on days 2 and 6 (*p* < 0.1). Tyr was higher in the degenerate group than in the growth group on day 6 (*p* < 0.05) and tended to be higher in the degenerate group than in the growth group on day 2 (*p* < 0.1). Hypro was higher in the degenerate group than in the growth group on day 2 (*p* < 0.05). 

### 3.3. Difference in Amino Acid Metabolism between Primordial Follicle Activation and Secondary Follicle Growth In Vitro

Interaction between days of culture and types of culture was demonstrated for 9 essential amino acids (Arg, His, Ile, Leu, Lys, Phe, Thr, Trp, and Val), 3 proteinogenic non-essential amino acids (Gly, GluNH_2_, and Pro), Hypro, EOHNH_2_, PEA, and Tau (*p* < 0.05) (Table 3). The culture period affected 7 essential amino acids (Arg, Ile, Leu, Lys, Phe, Thr, and Val), 4 proteinogenic non-essential amino acids (Gly, GluNH_2_, Pro, and Tyr), 3 non-proteinogenic amino acids (Hylys, Hypro, and Orn), EOHNH_2_, and PEA. The type of culture affected 4 essential amino acids (Ile, Leu, Lys, and Met), 10 proteinogenic non-essential amino acids (AspNH_2_, Asp, Ala, Cystine, Gly, GluNH_2_, Glu, Pro, Ser, and Tyr), 4 non-proteinogenic amino acids (Cit, Hylys, Hypro, and P-Ser), Urea, NH_3_, PEA, and Tau. 

As shown in Figure 7, 71.0% (22/31) of molecular species, including 7 essential amino acids (Arg, His, Met, Phe, Thr, Trp, and Val), 6 proteinogenic non-essential amino acids (Asp, Ala, Gly, Glu, Pro, and Tyr), 4 non-proteinogenic amino acids (β-Ala, Cit, Hypro, and Orn), and Urea, EOHNH_2_, NH_3_, PEA, and Tau were higher in cortical strip culture than in secondary follicle culture (Figure 7and Appendix A). Conversely, 19.4% (6/31) of molecular species, including 4 proteinogenic non-essential amino acids (AspNH_2_, Cystine, GluNH_2_, and Ser) and 2 non-proteinogenic amino acids (Hylys and P-Ser) were higher in secondary follicle culture than cortical strip culture (Figure 7and Appendix A). On the other hand, 9.7% (3/31) of molecular species, including 3 essential amino acids (Ile, Leu, and Lys), were higher in secondary follicle culture than in strip culture during the early culture period (Figure 7and Appendix A), but they became higher in cortical strip culture than secondary follicle culture during the late culture period due to the increase in their concentrations in cortical strip culture (Figure 9and Appendix A).

## 4. Discussion

Analysis of biofluids has been investigated as a non-invasive marker to assess oocyte competence in human-assisted reproduction [22], so the measurement of metabolic by-products in spent culture media has the potential to be a non-invasive marker of the health of follicles grown in vitro. The application of metabolomics to ovine-isolated secondary follicles using a spent culture medium has been attempted, but no clear conclusions could be drawn [23]. In this study, we monitored 19 molecular species during the culture period of ovarian cortical strips containing mainly primordial follicles and 6 molecular species during the culture of isolated secondary follicles. By comparing extreme samples in both types of cultures, we identified four molecular species that showed a difference in relation to the activation of primordial follicles and the growth of secondary follicles.

During primordial follicle activation, concentrations of the 3 essential amino acids (Arg, Lys, and Met) were significantly higher in the group that showed high activation of growth compared to the low activation group, while that of α-AAA, a metabolic intermediate in the catabolic process of Lys [24], was higher in the low activation group during the early culture period. These results seem counterintuitive, as they suggest that metabolic activity of these essential amino acids during the early cultural period is associated with suppression and not activation of primordial follicles. In most mammals, primordial follicle activation can be initiated in vitro within 48 h via disruption of the Hippo signaling pathway after fragmentation of ovarian tissue [25,26]. 

In the secondary follicle culture, three amino acids, His, Tyr, and Hypro, were higher in the degenerate group than in the growth group. Interestingly, His and Tyr are glucogenic amino acids [27]. His is catabolised to oxaloacetate, and Tyr is catabolized to fumarate and acetoacetate. These metabolites enter gluconeogenesis via the tricarboxylic acid cycle (TCA) cycle, while Hypro is an abundant constituent of collagen and elastin [28], which are the major components of the extracellular matrix (ECM) [29]. ECM, such as collagen, has important roles in folliculogenesis [30] and is altered in human ovarian cortical strips during culture [31]. Collagen (type IV) is known to be expressed in primordial to antral follicles in mice [32,33], humans [34,35,36], and cattle [37]. Higher concentrations of His, Met, Tyr, and Hypro in the degenerate group than in the growth group may indicate that active metabolism and/or catabolism of these amino acids are signatures of healthy growth of secondary follicles. However, given that the focus was on extreme examples in each culture group and the sample sizes were small, it is clear that more focused studies with increased replicates and analysis of intermediate samples are required. Furthermore, given that the changes observed in amino acids are associated with the early stages of culture (day 2), it would be important to look at the expression of amino acid metabolism genes at different stages of follicle development and during the culture period.

Interestingly, the concentration of Met was higher in the high activation group in cortical strip culture but tended to be lower in the growth group in secondary follicle culture. Primordial follicle activation is triggered by the inactivation of some signaling pathways, such as PTEN [19] and Hippo signaling [25]. On the other hand, transcription and protein synthesis are active in growing follicles [8]. The difference between primordial follicle activation and secondary follicle growth in Met concentration may indicate the necessity of Met to maintain primordial follicle dormancy rather than activation of follicular growth.

Despite the limitations of this study, it does give indications of where to focus for future work. In addition to the differences observed within each culture system, the results also show differences in concentrations of all molecular species between cortical strip culture and secondary follicle culture, indicating that amino acid requirements are different between these two systems. In established two-step culture systems for primordial follicle activation and subsequent secondary follicular growth, the same basic medium is used at each stage [5,6,7,38,39]. Given the results observed in this study, it may be possible to optimize the basic media for each culture step in terms of amino acid metabolism and thus improve oocyte quality from follicles grown in a multi-step system.

## 5. Conclusions

In summary, this preliminary study has demonstrated differences in Arg, Lys, and Met, and α-AAA, a metabolite of Lys, is associated with primordial follicle activation in vitro. The results suggest that if the metabolism of these essential amino acids is active during the early culture period, activation of primordial follicles may be suppressed. In secondary follicle culture, His, Tyr, and Hypro were higher in the degenerate group than the growth group, indicating that secondary follicles with higher growth competence can metabolize or catabolize those amino acids actively. Taken together, we conclude that amino acid metabolism could be used as an indicator of primordial follicle activation and subsequent growth competence in bovine IVG. However, further detailed studies are required.

## Figures and Tables

**Figure 1 animals-13-01141-f001:**
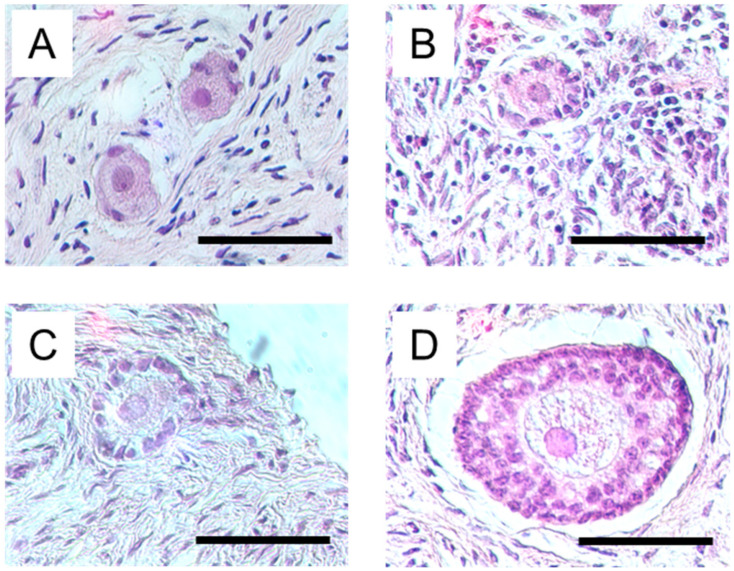
Photomicrographs of histological sections of follicles at each stage of development. (**A**): Primordial follicles: an oocyte surrounded by a complete or incomplete single layer of a flattened granulosa cell. (**B**): Transitional follicles: oocyte surrounded by a mixed layer of flattened and cuboidal granulosa cells. (**C**): Primary follicles: oocyte surrounded by a single layer of cuboidal granulosa cells. (**D**): Secondary stage: oocyte surrounded by two or more complete layers of cuboidal granulosa cells. Scale bar = 50 µm.

**Figure 2 animals-13-01141-f002:**
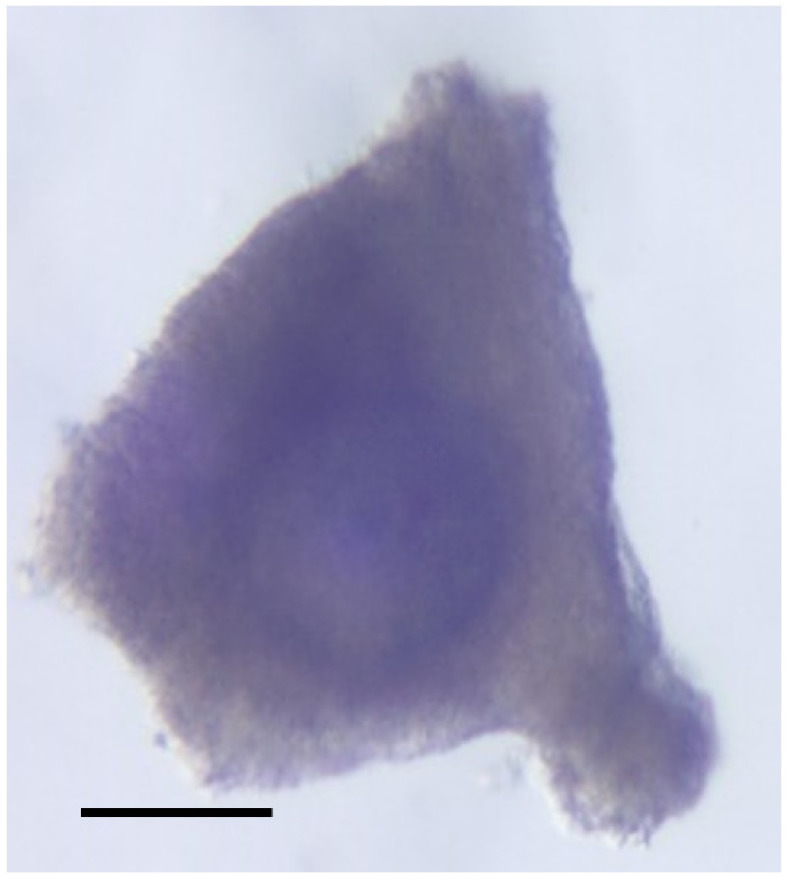
Isolated secondary follicle. Secondary follicles (100–200 μm) were mechanically dissected, and those with an intact basement membrane and no antral cavity were selected for culture. Scale bar = 100 µm.

**Figure 3 animals-13-01141-f003:**
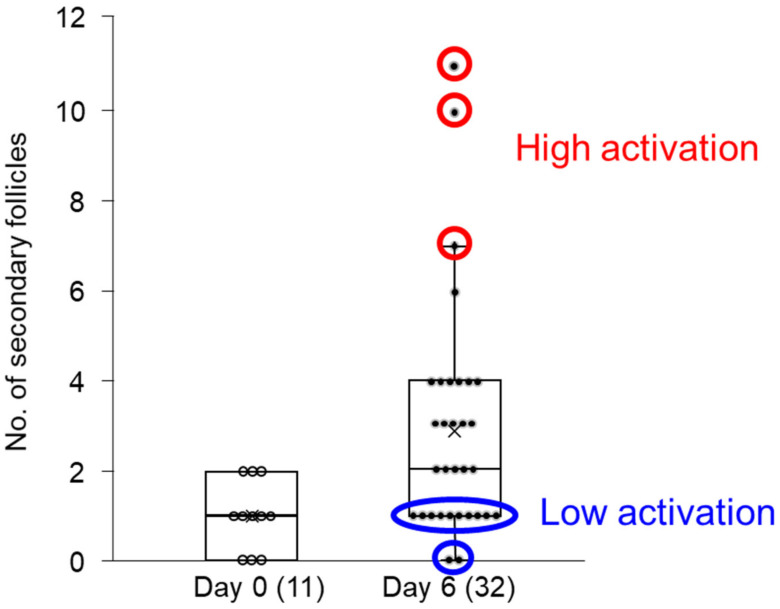
Distribution of the number of secondary follicles on days 0 and 6 of culture. Lines on the boxes delineate the 25th, 50th, and 75th percentiles, whereas the whiskers depict the minimum values or 1.5 times the interquartile range. X markers mean averages. Samples selected as the high activation group were marked by a red circle, while samples selected as the low activation group were marked by a blue circle.

**Figure 4 animals-13-01141-f004:**
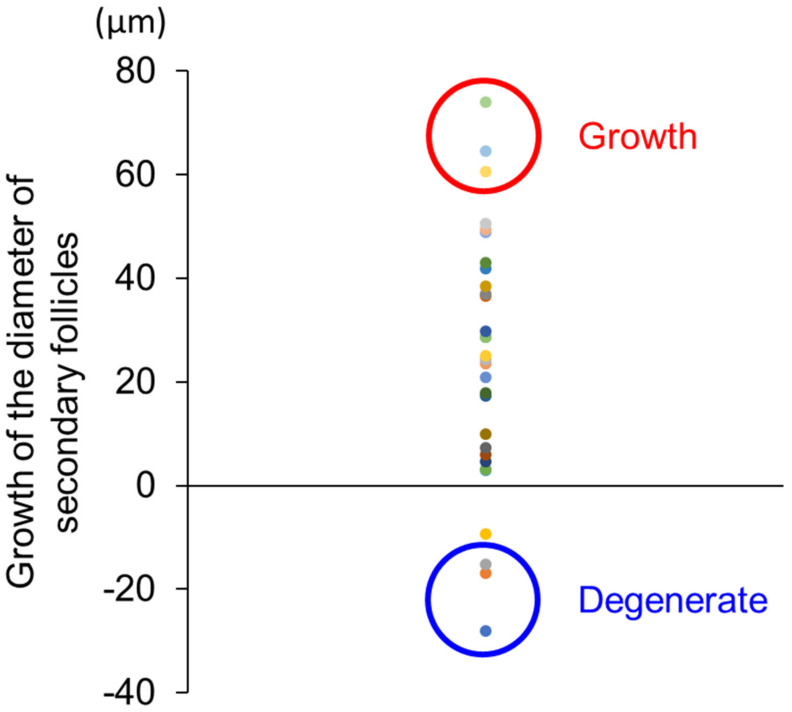
Distribution of the growth of the diameter of secondary follicles during culture. Growth of secondary follicles was expressed as the difference in the diameter of each follicle on days 0 and 6. Samples selected as the growth group were marked by a red circle, while samples selected as the degenerate group were marked by a blue circle.

**Figure 5 animals-13-01141-f005:**
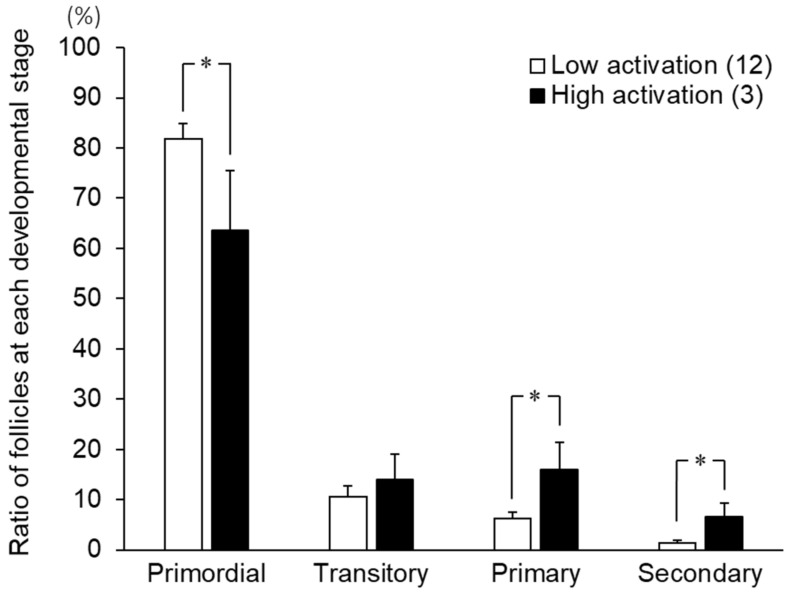
The ratio of follicles at each developmental stage of the low and high activation groups on day 6 of culture. * An asterisk indicates a significant difference between the low and high activation groups (*p* < 0.05). Numbers in parentheses indicate the number of strips in each group. Error bars indicate the standard error of the mean (SEM).

**Figure 6 animals-13-01141-f006:**
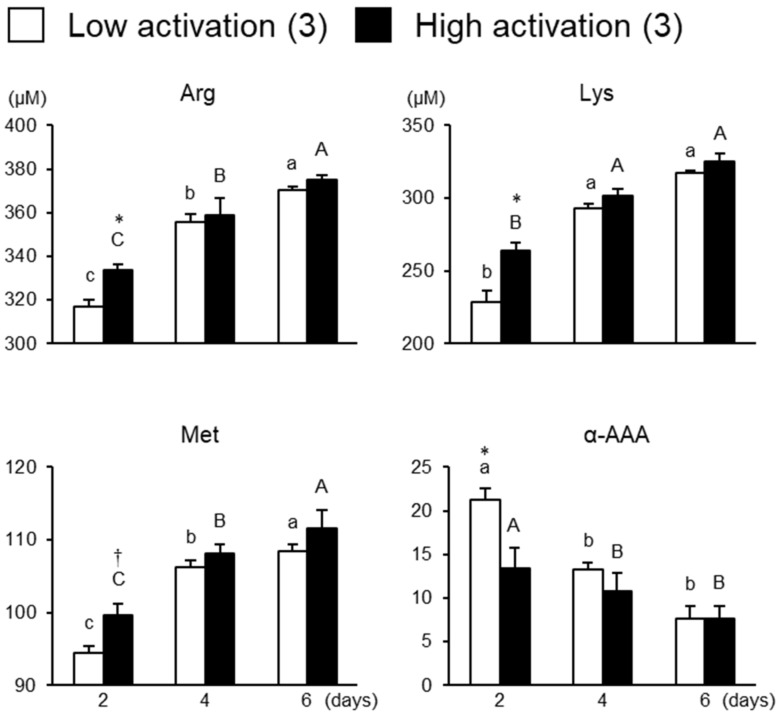
Concentrations of the molecular species that showed differences between the low and high activation groups in culture media. a–c: Different letters indicate significant differences between different cultural periods in the low activation group (*p* < 0.05). A–C: Different letters indicate significant differences between different cultural periods in the high activation group (*p* < 0.05). * An asterisk indicates a significant difference between the low and high activation groups (*p* < 0.05). † A dagger indicates a tendency in the difference between the low and high activation groups (*p* < 0.1). Numbers in parentheses indicate the number of the strip in each group. Error bars indicate SEM.

**Figure 7 animals-13-01141-f007:**
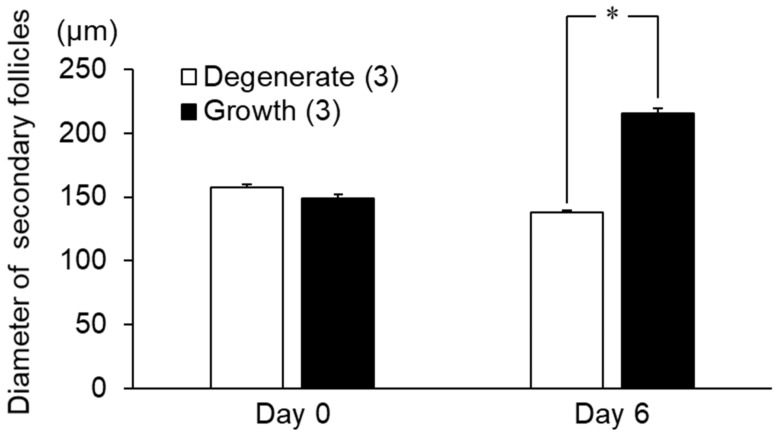
The diameter of secondary follicles was selected for the free amino acid analysis as the growth and degenerate groups before and after secondary follicle culture. * An asterisk indicates a significant difference between the growth and degenerate groups (*p* < 0.05). Numbers in parentheses indicate the number of follicles in each group.

**Figure 8 animals-13-01141-f008:**
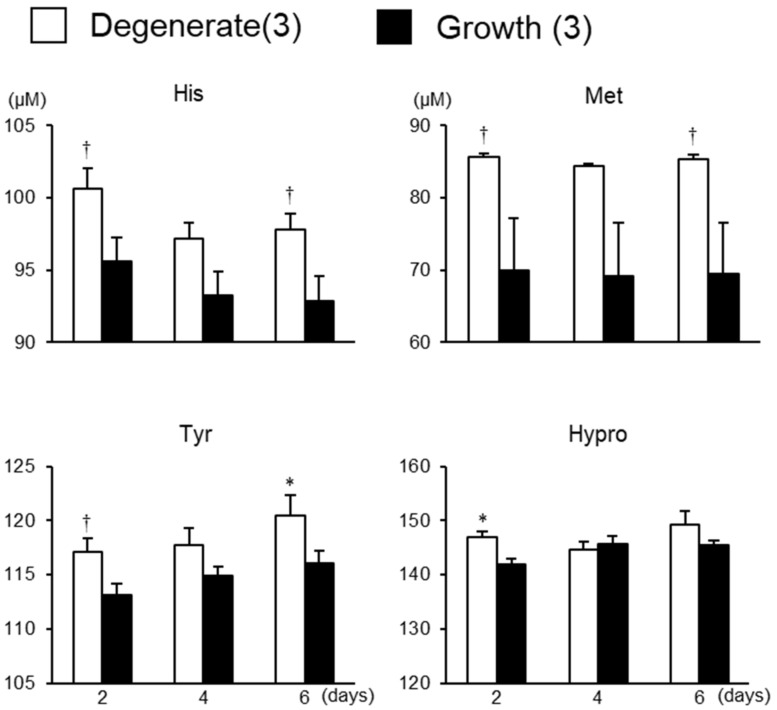
Concentrations of the molecular species that showed differences between the growth and degenerate groups in culture media. * An asterisk indicates a significant difference between the growth and degenerate groups (*p* < 0.05). † A dagger indicates a tendency in the difference between the growth and degenerate groups (*p* < 0.1). Numbers in parentheses indicate the number of follicles in each group. Error bars indicate SEM.

**Figure 9 animals-13-01141-f009:**
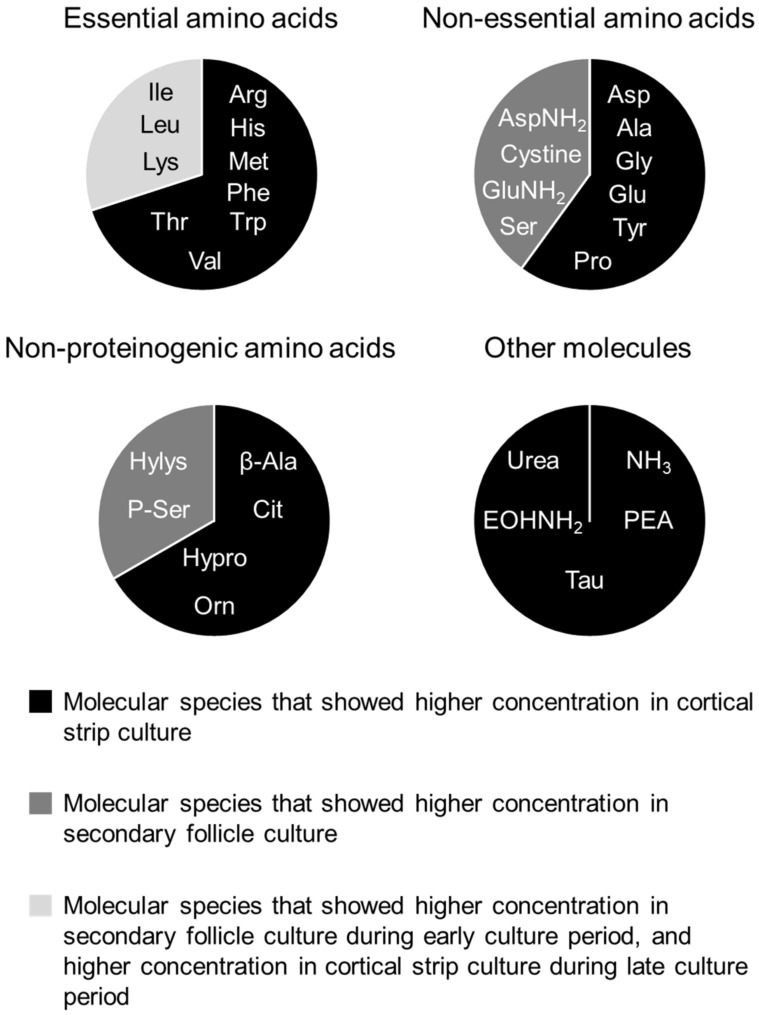
The molecular species that showed differences between the cortical strip culture and secondary follicle cultures. All molecules detected by the free amino acid analysis were categorized as essential amino acids, non-essential amino acids, non-proteinogenic amino acids, and other molecules. Concentrations of these molecules were compared between cortical strip culture for primordial activation and secondary follicle culture. Molecules in the black area were higher in cortical strip culture than in secondary follicle culture throughout the culture period. Molecules in the dark gray area were higher in secondary follicle culture than in cortical strip culture. Molecules in the light area were higher in secondary follicle culture than in cortical strip culture during the early culture period, then became higher in fragment culture than in secondary follicle culture during the late culture period.

**Table 1 animals-13-01141-t001:** *p*-values associated with the groups (low activation vs. high activation) and culture periods (day 2 vs. day 4 vs. day 6) in cortical strip culture of bovine ovaries on amino acid metabolism.

Type of Parameters	Parameters	*p*-Value		
		Day × Group	Day	Group
Essential amino acids	Arg	NS	**<0.0001**	**0.0102**
	His	NS	**0.0112**	NS
	Ile	NS	**0.0023**	0.0971
	Leu	NS	**0.0003**	NS
	Lys	**0.0379**	**<0.0001**	**0.0005**
	Met	NS	**<0.0001**	**0.0316**
	Phe	NS	**<0.0001**	0.0952
	Thr	NS	**0.0046**	NS
	Trp	NS	**0.0329**	NS
	Val	NS	**0.0009**	NS
Non-essential amino acids	AspNH_2_	NS	NS	NS
	Asp	NS	NS	NS
	Ala	NS	NS	NS
	Cystine	NS	NS	NS
	Gly	NS	**0.0013**	NS
	GluNH_2_	NS	NS	NS
	Glu	NS	NS	NS
	Pro	NS	**0.0259**	NS
	Ser	NS	NS	NS
	Tyr	NS	**0.0290**	0.0823
Non-proteinogenic amino acids	α-AAA	0.0965	**0.0003**	**0.0061**
	β-Ala	NS	NS	NS
	Cit	NS	NS	NS
	Hylys	NS	**0.0066**	NS
	Hypro	NS	**0.0117**	NS
	Orn	NS	**0.0067**	NS
	P-Ser	NS	NS	NS
Other molecules	Ans	NS	NS	NS
	Urea	NS	NS	NS
	EOHNH_2_	NS	NS	NS
	NH_3_	NS	NS	NS
	PEA	NS	**0.0453**	NS
	Tau	NS	0.0607	NS

Values in bold—significant difference (*p* < 0.05), NS—no significant difference (*p* > 0.1).

**Table 2 animals-13-01141-t002:** *p*-values associated with the groups (growth vs. degenerate) and culture periods (day 2 vs. day 4 vs. day 6) in bovine secondary follicle culture on amino acid metabolism.

Type of Parameters	Parameters	*p*-Value		
		Day × Group	Day	Group
Essential amino acids	Arg	NS	NS	NS
	His	NS	NS	**0.0334**
	Ile	NS	NS	NS
	Leu	NS	NS	NS
	Lys	NS	NS	NS
	Met	NS	NS	0.0528
	Phe	NS	NS	NS
	Thr	NS	NS	NS
	Trp	NS	NS	NS
	Val	NS	NS	NS
Non-essential amino acids	AspNH_2_	NS	NS	NS
	Asp	NS	**0.0061**	NS
	Ala	NS	0.0613	NS
	Cystine	NS	0.0637	NS
	Gly	NS	**0.0322**	NS
	GluNH_2_	NS	**0.0010**	0.094
	Glu	NS	NS	NS
	Pro	NS	NS	NS
	Ser	NS	NS	NS
	Tyr	NS	0.0956	0.0580
Non-proteinogenic amino acids	β-Ala	NS	NS	NS
	Cit	NS	NS	NS
	Hylys	NS	NS	NS
	Hypro	NS	NS	**0.0362**
	Orn	NS	NS	NS
	P-Ser	NS	**0.0020**	NS
Other molecules	Urea	NS	**0.0060**	NS
	EOHNH_2_	NS	NS	NS
	NH_3_	NS	**0.0020**	NS
	PEA	**0.0373**	NS	NS
	Tau	NS	NS	NS

Values in bold indicate—significant difference (*p* < 0.05), NS—no significant difference (*p* > 0.1).

**Table 3 animals-13-01141-t003:** *p*-values associated with the types of culture (primordial vs. secondary) and culture periods (day 2 vs. day 4 vs. day 6) on amino acid metabolism.

Type of Parameters	Parameters	*p* Value		
		Day × Type of Culture	Day	Type of Culture
Essential amino acids	Arg	**<0.0001**	**<0.0001**	NS
	His	**0.0033**	NS	NS
	Ile	**0.0035**	**0.0025**	**0.0422**
	Leu	**0.0002**	**<0.0001**	**0.0045**
	Lys	**<0.0001**	**<0.0001**	**<0.0001**
	Met	NS	NS	**0.0005**
	Phe	**<0.0001**	**<0.0001**	0.0666
	Thr	**0.0007**	**0.0016**	NS
	Trp	**0.032**	0.0742	NS
	Val	**0.0005**	**0.0001**	0.0632
Non-essential amino acids	AspNH_2_	NS	NS	**<0.0001**
	Asp	NS	NS	**<0.0001**
	Ala	NS	NS	**<0.0001**
	Cystine	NS	NS	**0.0092**
	Gly	**0.0002**	**<0.0001**	**<0.0001**
	GluNH_2_	**0.0177**	**0.0028**	**0.0131**
	Glu	NS	NS	**<0.0001**
	Pro	**0.0144**	**0.0051**	**0.0001**
	Ser	NS	NS	**<0.0001**
	Tyr	NS	**0.013**	**0.0001**
Non-proteinogenic amino acids	β-Ala	NS	NS	0.0627
	Cit	NS	NS	**<0.0001**
	Hylys	NS	**0.0044**	**0.0156**
	Hypro	**0.0089**	**0.0013**	**0.0006**
	Orn	0.053	**0.0007**	NS
	P-Ser	NS	NS	**0.0010**
Other molecules	Urea	NS	NS	**0.0143**
	EOHNH_2_	**<0.0001**	**<0.0001**	NS
	NH_3_	NS	0.0508	**0.0007**
	PEA	**0.0195**	**0.0261**	**<0.0001**
	Tau	**0.0248**	NS	**<0.0001**

Values in bold indicate—significant difference (*p* < 0.05), NS—no significant difference (*p* > 0.1).

## Data Availability

The datasets used and/or analyzed during the present study are available from the corresponding authors upon reasonable request.

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
