# Peer review of "Relationship between Amino Acid Metabolism and Bovine In Vitro Follicle Activation and Growth"

_animals, 2023, doi:10.3390/ani13071141_

Round 1

Reviewer 1 Report

This study evaluated the amino acid metabolism of bovine follicles during in vitro growth (IVG) to identify indicators associated with follicle survival and growth during in vitro culture. The authors have performed extensive experiments showing that the profile of amino acid metabolism in spent culture media could be associated with the growth or healthy state of in vitro cultured bovine follicles.

Although the written language standard is globally acceptable; the general flow of the text needs to be improved.

The discussion is the weakest part of the manuscript, both in its content, structure, and English. It is difficult to read. The authors do compare their results to those in the literature, but they do not present clear interpretations of the data, nor do they use them to elaborate clear hypotheses.

Comments by section

Abstract

Line 24: Was the tissue examined histologically after 6 days of culture?

Introduction

I would suggest describing what primordial follicle activation is.

Material and method

What is primordial follicle activation, define it in the introduction section

Line 188: What is fragment culture? cortex slices?, define or use the same definition throughout the article

Are there any other studies in which follicles have been classified as Growth or Degeneration based on the change in their diameter during in vitro culture? If yes, please add the reference(s).

Has it previously been suggested that the change in the diameter of follicles during in vitro culture is associated with growth or degeneration? If yes, please add the reference(s).

Results

Why include trends in the results? Do the trends contribute to the interpretation of the results in the discussion section?

Discussion

I suggest restructuring the discussion section as follows

First paragraph briefly conclusion of the results and meaning with respect to what is already known from previous studies.

Do not repeat results please (i.e. lines 390-393; 400-402)

You don't want to go into too much detail or repeat yourself by describing your results again. Rather, a quick transition to what the results mean and explain their impact.

Lines 403-410: all this description of each aminoacid seems to be not necessary and does not seem to contribute to the interpretation of the results

Lines 418-419: the same, this sentence does not seem to lead to the discussion, but rather could be part of the introduction

Lines 424-429: I suggest including this in the discussion as a conclusion

Author Response

We thank the reviewers for their positive comments and helpful suggestions. We have revised the manuscript to take into account these suggestions. We have responded to each reviewer in turn and have outlined how we have dealt with these comments in our revised manuscript.

Reviewer 1

This study evaluated the amino acid metabolism of bovine follicles during in vitro growth (IVG) to identify indicators associated with follicle survival and growth during in vitro culture. The authors have performed extensive experiments showing that the profile of amino acid metabolism in spent culture media could be associated with the growth or healthy state of in vitro cultured bovine follicles.

Although the written language standard is globally acceptable; the general flow of the text needs to be improved.

The manuscript has been substantially edited to improve the flow and language.

The discussion is the weakest part of the manuscript, both in its content, structure, and English. It is difficult to read. The authors do compare their results to those in the literature, but they do not present clear interpretations of the data, nor do they use them to elaborate clear hypotheses.

The discussion has been rewritten with these suggestions incorporated in the current L389-441.

Comments by section

Abstract

Line 24: Was the tissue examined histologically after 6 days of culture?

 Yes, tissue was examined histologically with samples taken before culture and at the end of 6 days in culture. This has been clarified in the abstract in the current L24.

Introduction

 I would suggest describing what primordial follicle activation is.

 This has been clarified in the introduction in the current L47-48.

Material and method

 What is primordial follicle activation, define it in the introduction section

This has been defined in the current L47-48.

Line 188: What is fragment culture? cortex slices?, define or use the same definition throughout the article

Fragment culture has been changed to cortical strip culture throughout to describe strips of ovarian cortex of similar dimensions that are prepared for culture to support the activation of primordial follicle growth.

Are there any other studies in which follicles have been classified as Growth or Degeneration based on the change in their diameter during in vitro culture? If yes, please add the reference(s).

It has previously been suggested that the change in the diameter of follicles during in vitro culture is associated with growth and degeneration in isolated bovine secondary follicles. A reference has been added in L220.

Results 

Why include trends in the results? Do the trends contribute to the interpretation of the results in the discussion section?

We agree that the discussion of trends is vague and this has been removed from the revised version from the current L408-411 (discussion) and L447 (conclusion). The focus is now on significant differences only. 

Discussion

I suggest restructuring the discussion section as follows

First paragraph briefly conclusion of the results and meaning with respect to what is already known from previous studies.

Do not repeat results please (i.e. lines 390-393; 400-402)

You don't want to go into too much detail or repeat yourself by describing your results again. Rather, a quick transition to what the results mean and explain their impact.

Lines 403-410: all this description of each amino acid seems to be not necessary and does not seem to contribute to the interpretation of the results

Lines 418-419: the same, this sentence does not seem to lead to the discussion, but rather could be part of the introduction

Lines 424-429: I suggest including this in the discussion as a conclusion

Thank you for your suggestions. The discussion has been rewritten and restructured to be more succinct and focused in the current L389-441.

Reviewer 2 Report

The Kenichiro et al., manuscript ID: animals-2219914 evaluated amino acid metabolism of bovine follicles during in vitro growth to identify indicators of health status of follicles during culture. It is interest to study the amino acid metabolism of follicles in vitro so as to select the appropriate amino acid concentration for follicle culture. There are several suggestions for this manuscript.

1. The composition of the culture medium directly affects cellular metabolism. The authors used McCoy's 5a medium as the basal culture medium for fragment and secondary follicle culture. However, McCoy's 5a medium contains more than 20 kinds of amino acid components, including Lys, Met, etc., and it is worth evaluation whether the differentiated or redundant amino acid content in this medium will affect its amino acid metabolism. Did the authors evaluate differences in other types of media?

2. The authors claiming that Met was higher in the High activation fragment, but was lower in the Growth follicle. It is well known that the activation of follicles requires the growth of follicles, which seems to contradict for the two results. The author needs to explain the reasons for this difference.

3. Most of the significant changes in amino acids in the manuscript mainly appeared in the early stage of follicle development (2 day), and it is worth verifying whether the utilization of amino acids in follicles is mainly in the recruitment stage rather than the later stage. The authors should demonstrate this by adding the expression of amino acid metabolism genes during early and late follicular development.

4. The up-regulation or down-regulation of amino acids in different groups cannot be seen in Table 1 and 2, so it is very necessary to show specific amino acid concentration values in the table.

Author Response

We thank the reviewers for their positive comments and helpful suggestions. We have revised the manuscript to take into account these suggestions. We have responded to each reviewer in turn and have outlined how we have dealt with these comments in our revised manuscript.

Reviewer 2

The Kenichiro et al., manuscript ID: animals-2219914 evaluated amino acid metabolism of bovine follicles during in vitro growth to identify indicators of health status of follicles during culture. It is interest to study the amino acid metabolism of follicles in vitro so as to select the appropriate amino acid concentration for follicle culture. There are several suggestions for this manuscript.

Thank you

  1. The composition of the culture medium directly affects cellular metabolism. The authors used McCoy's 5a medium as the basal culture medium for fragment and secondary follicle culture. However, McCoy's 5a medium contains more than 20 kinds of amino acid components, including Lys, Met, etc., and it is worth evaluation whether the differentiated or redundant amino acid content in this medium will affect its amino acid metabolism.Did the authors evaluate differences in other types of media?

This is a good suggestion but in this study we focused on the basic media that is used in the culture system that we have worked on.

  1. The authors claiming that Met was higher in the High activationfragment, but was lower in the Growth follicle. It is well known that the activation of follicles requires the growth of follicles, which seems to contradict for the two results. The author needs to explain the reasons for this difference.

Thank you for this interesting observation. We have discussed these differences in Met concentration between Experiment 1 (primordial follicle activation) and Experiment 2 (secondary follicle growth) in more detail according to your comment L425-441. These differences may reflect the process of regulation of activation versus follicle growth processes.

  1. Most of the significant changes in amino acids in the manuscript mainly appeared in the early stage of follicle development (2 day), and it is worth verifying whether the utilization of amino acids in follicles is mainly in the recruitment stage rather than the later stage. The authors should demonstrate this by adding the expression of amino acid metabolism genes during early and late follicular development.

We agree that this should be done but it is out with the scope of this preliminary study and has been highlighted in the manuscript in the current L421-421 as future work.

  1. The up-regulation or down-regulation of amino acids in different groups cannot be seen in Table 1 and 2, so it is very necessary to show specific amino acid concentration values in the table.

This is shown in the supplementary information which is referred to in the text. We stated as “All concentrations are shown in Supplementary Materials” in the current Line 275-276.

Reviewer 3 Report

In the present study, the authors conducted three kinds of studies: (1) They cultured ovarian fragments to study amino acid metabolism; (2) They isolated secondary follicles to investigate amino acid metabolism; (3) They compared the amino acid metabolism between fragment culture and secondary follicles. The authors intended to find relationship between amino acid metabolism and early follicle development, and identify markers for follicular health. This interesting paper provides a novel insight into the role of metabolism on bovine follicular development. The data are of high quality and the paper is clearly written and well reasoned.

Major concern:

The results of figure 5 and figure 6 were from different raw data, i.e., all the Low activation group were used to calculate the ratio of follicular stages, but only one of them was selected for the amino acid analysis. Will this cause misleading results? The differences in Primordial, Primary and Secondary may not be significant after excluding the remaining nine samples in the Low activation group.

Minor:

Line 121,”Transitional follicles in the image and legend.

Author Response

We thank the reviewers for their positive comments and helpful suggestions. We have revised the manuscript to take into account these suggestions. We have responded to each reviewer in turn and have outlined how we have dealt with these comments in our revised manuscript.

Reviewer 3

Comments and Suggestions for Authors

In the present study, the authors conducted three kinds of studies: (1) They cultured ovarian fragments to study amino acid metabolism; (2) They isolated secondary follicles to investigate amino acid metabolism; (3) They compared the amino acid metabolism between fragment culture and secondary follicles. The authors intended to find relationship between amino acid metabolism and early follicle development, and identify markers for follicular health. This interesting paper provides a novel insight into the role of metabolism on bovine follicular development. The data are of high quality and the paper is clearly written and well reasoned.

Thank you for these kind comments.

Major concern:

The results of figure 5 and figure 6 were from different raw data, i.e., all the Low activation group were used to calculate the ratio of follicular stages, but only one of them was selected for the amino acid analysis. Will this cause misleading results? The differences in Primordial, Primary and Secondary may not be significant after excluding the remaining nine samples in the Low activation group.

Three samples that showed extreme low activation were selected for amino acid analysis as three samples were selected that showed the highest rate of activation. The differences between these samples remain in terms of activation rate.

Minor:

Line 121,”Transitional follicles” in the image and legend.

This has been corrected
